# A synthetic control evaluation of the use of pulse oximeters in response to the COVID-19 pandemic in England

Stefano Conti[1,2]*, Paris Pariza[3], Arne Wolters[4]

1 Improvement Analytics Unit, the Health Foundation, London, United Kingdom, 2 Modelling and System Analytics, Data & Analytics, NHS England, London, United Kingdom, 3 Clinical and Improvement Analytics, NHS England, London, United Kingdom, 4 Improvement Analytics, NIHR Applied Research Collaboration for the North East and North Cumbria, Cumbria, Northumberland, Tyne & Wear NHS Foundation Trust, Newcastle upon Tyne, United Kingdom

* stefano.conti@nhs.net

## Abstract

### Objectives

To measure the impact of the use of pulse oximeters in early detection of oxygen saturation deterioration for patients testing positive with COVID-19 in preventing emergency hospital utilisation and death. This intervention was rolled out across England (part of the United Kingdom) as the COVID Oximetry @home programme.

### Design

Causal inference study informed by linked national administrative and surveillance data-sets to detect the difference in impact of the COVID Oximetry @home programme by comparing areas of the country with high uptake of the programme with areas with low uptake of the programme using generalised synthetic controls.

### Setting

This intervention was rolled out across all Clinical Commissioning Groups (CCGs) in England, administrative geographical areas often aligned with local authorities. All CCGs were invited to submit participation data to the National Health Service in England. Those CCGs that submitted complete data were included in the study.

### Participants

Patients registered with participating CCGs who tested positive for COVID-19, and where either 65 years of age or over, or clinically extremely vulnerable.

### Outcomes

A&E attendances, emergency admissions, admissions into critical care and mortality within 28 days of a positive COVID-19 diagnosis.

**Data availability statement:** Data informing this observational study are unavailable due to the Control Of Patient Information (COPI) notice issued by NHS Digital having expired on 30 June 2022 (URL: https://www.england.nhs.uk/wp-content/uploads/2022/07/C1639_ii-co-pi-notice-expiration-on-30-june-22.pdf). COPI Notices provided us with the necessary and sufficient legal basis for processing the confidential patient information underpinning our evaluation that was required, at the time of its implementation, for managing the UK response to the COVID-19 pandemic. Upon expiration of the COPI Notices the UK Department of Health and Social Care deemed said legal basis to be no longer sustainable for the purpose of managing the response to the COVID-19 pandemic. Accordingly the full data-set informing our evaluation, which included and was linked to confidential patient information obtained under the expired COPI Notices, had to be contextually deleted from the secure data environment it was stored in, and made accessible to, the Improvement Analytics Unit. In the event of an alternative lawful basis being identified that would permit continued processing of confidential patient information for COVID-19 purposes after 30 June 2022 – based for example on obtaining patient consent, on Regulation 3 of the Health Service COPI Regulations 2002 or on transitioning to Regulation 5 of the Health Service COPI Regulations 2002 – applicants can contact the NHS Health Research Authority Confidentiality Advisory Group (URL: https://www.hra.nhs.uk/about-us/committees-and-services/confidentiality-advisory-group/; e-mail: cag@hra.nhs.uk): an independent body providing expert advice on the use of confidential patient information whose mission is to protect and promote the interests of patients and the public in the UK, while at the same time facilitating appropriate use of confidential patient information for purposes beyond direct patient care.

**Funding:** The author(s) received no specific funding for this work.

**Competing interests:** The authors have declared that no competing interests exist.

## Results

No differences were detected in the rate of emergency hospital use or mortality between CCGs with high uptake and CCGs with low uptake.

## Conclusion

The lack of impact detected on all outcomes of interest may simply be due to an absence of impact. Factors that may have impacted the ability to detect an effect are the low uptake of the programme, heterogeneity in the implementation of the pathway, or design limitations of the study.

## Introduction

As part of the response to the sustained spread of the COVID-19 pandemic in England during Winter 2020−21, the National Health Service (NHS) began rolling out pulse oximeters in support of remote monitoring of falling oxygen blood levels among people testing positive for the infection. The first wave of the pandemic saw a peak in hospital admissions among patients diagnosed with COVID-19 suffering from 'silent' hypoxia (i.e., asymptomatic presentation with low blood oxygen saturation), who would subsequently experience extended hospital stays, invasive ventilation, intensive care treatment and fatality [1]. The NHS set out to detect at an early stage cases of rapidly deteriorating patients with COVID-19 through remote (typically from their usual place of residence) pulse oximetry, with the aims of (i) avoiding unnecessary hospital admissions and mortality and (ii) escalating deteriorating cases to urgent, non-critical care, in the spirit of the "appropriate care at the appropriate place" NHS campaign (URL: https://www.england.nhs.uk/blog/giving-care-in-the-right-place-first-time/). Service implementation was initially piloted during the first wave of the pandemic [2] and scaled up to a national programme (COVID Oximetry @home, or CO@h) in November 2020. By the end of January 2021 all Clinical Commissioning Groups (CCGs) in England were set up for provision of a CO@h clinical pathway. In practice, there was a lot of variation over time as well as between sites in implementation of CO@h pathways, with differences including variations in the point of service (pre-hospital or post-discharge), models (primary care, secondary care, step-down or mixed) and, crucially, eligibility criteria [3].

Evidence around the effectiveness and cost-effectiveness of pulse oximetry in England is limited and developing, likely dependent on the implementation context and clinical settings. Studies around the use of pulse oximetry for remote monitoring in England during the first wave of the pandemic focussed largely on its safety and implementation models [4–5]. A consortium of UK academic and research organisations – comprising the National Institute for Health and Care Research's (NIHR) Rapid Service Evaluation Team (RSET) and Birmingham, RAND and Cambridge Evaluation (BRACE); NIHR's and Imperial College London's Patient Safety Translational Research Centre (PSTRC); and the Health Foundation's and NHS England and NHS Improvement's Improvement Analytics Unit (IAU) – was formed in Autumn

2020 with the purpose of filling the evidence gap surrounding the CO@h programme. This study illustrates findings from a population-level evaluation of the impact of the CO@h programme on emergency hospital activity and mortality, based on a variety of linked data sources from primary and secondary care, COVID-19 surveillance and mortality statistics. The conclusions it reaches, which are outlined in the Results section and interpreted in the Discussion section, corroborate, and should be framed in the wider context of, evidence appraisals independently contributed to by partnering teams in the consortium [6].

## Methods

According to guidelines issued by the NHS Health Research Authority (URL: https://www.hra-decisiontools.org.uk/research/docs/definingresearchtable_oct2017-1.pdf), since the submitted manuscript describes a service evaluation it requires no NHS Research Ethics Committee review.

Additionally, on April the 1st 2020 the United Kingdom's Secretary of State for Health and Social Care issued Notices under Regulation 3(4) of the Health Service (Control of Patient Information) Regulations 2002 (COPI), which required organisations to share confidential patient information with organisations – including the Improvement Analytics Unit, which all authors were part of at the time of this study being carried out – entitled to process this under COPI for COVID-19 purposes (COPI Notices). We have received anonymised patient-level data to inform the impact evaluation outlined in this submission under the above COPI Notices, which have provided us with the necessary and sufficient legal basis for the processing of the confidential patient information underpinning our evaluation that was required for managing the response to the COVID-19 pandemic; said COPI Notices have expired of June the 22nd 2022.

As explained in the Introduction, the roll-out of pulse oximeters within the CO@h programme did not take place evenly and simultaneously across England, but was staggered across sites (GP practices, A&E departments, community care teams, NHS trusts) and CCGs since the start of the pandemic. Although the national CO@h programme commenced in December 2020, a significant and growing number of local health services across the country have been providing their patients at risk of health deterioration due to COVID-19 with pulse oximeters since May 2020. In particular national coverage of the CO@h programme during Winter 2020−21 increased sharply: according to submission reports collected by the Kent, Surrey and Sussex Academic Health Science Network (AHSN) [7], the number of CCGs in England offering the CO@h pathway to their patients grew from 13 to 80 and 130 respectively by the end of December 2020 and January 2021.

Such an uneven but steady programme adoption pattern proved especially challenging when designing an impact evaluation, even more so around the introduction of a health-care intervention at a time the health-care system was under unprecedent and sustained pressure. Due to the absence of a natural comparison group, the evaluation strategy that was adopted focussed on differences in emergency hospital use and mortality between CCGs offering the programme to either a high or low proportion of eligible patients based on the 'CO@h onboarding fraction'. Under this premise, findings from such evaluation might be expected to be potentially sensitive to the thresholds used to classify a CCG as high- or low-onboarding. As such, the thresholding applied to the CO@h onboarding fraction was subjected to sensitivity analysis.

The evaluation thus examined emergency hospital use and COVID-19 mortality during the second wave of the pandemic among adults who, per the programme's eligibility criteria [3], tested positive to COVID-19 and were either of 65 years of age or older, or classed as clinically extremely vulnerable [8], and were registered with a GP practice in any CCG in England participating into the CO@h programme. Emergency hospital use was quantified through aggregate (CCG-level) rates of occurrence, within 28 days of a first positive COVID-19 test, of: A&E attendance; emergency admission; admission into critical care. Sparsity in the data collected from the programme on length of hospital stay (measured as overnight bed days) hampered successful modelling of this outcome metric.

In order to gauge the effectiveness of the CO@h programme among high-onboarding CCGs relative to low-onboarding areas, emergency hospital activity and mortality among the former needed estimating under a low-onboarding scenario

throughout the study follow-up period (the 'counterfactual'). The impact of the CO@h programme was estimated by comparing longitudinal observed outcomes from high-onboarding CCGs and their corresponding counterfactual. A counterfactual was separately modelled for each outcome by using the Generalized Synthetic Control (GSynth) approach [9], which fits a linear interactive fixed effects regression model [10] to data available on the intervention (before the follow-up period only) and control (before and throughout the follow-up period) CCGs to predict longitudinal outcome trajectories that high-onboarding CCGs would have shown, had they instead been onboarding low proportions of eligible patients. Compared to other common counterfactual-building strategies, the GSynth method was appealing for the present analysis in that is applicable to multiple intervention units, accounts for unobservable time-varying effects (if present), detects the presence of unobserved confounding (hidden bias) and produces coherent inferences around impact parameters [11]. Data processing and analysis were carried out in the Secure Data Environment managed by The Health Foundation using the R statistical programming language [12]; impact estimation via GSynth relied on facilities encoded in its 'gsynth' library [13]. Proposed GSynth models were satisfactorily validated via a number of diagnostic statistics, including placebo tests, visual inspection of estimated latent factors and corresponding loadings, mean squared prediction error and Bayesian Information Criterion statistics.

Data informing the evaluation were obtained from an array of administrative sources. Baseline characteristics on patients registered in a GP practice in England were derived from the General Practice Extraction Service Data for Pandemic Planning and Research [14]. Testing data were obtained from the Second Generation Surveillance System [15], which collects COVID-19 test results from laboratories across England. For patients with multiple tests on record only the date of the earliest COVID-19 test was retained. Onboarding data on enrolled patients were submitted from participating sites via NHS Digital's Strategic Data Collection Service [16]. Secondary care data on A&E attendances and emergency admissions of onboarded patients were sourced from NHS Digital's Hospital Episode Statistics [17] and the Emergency Care Data Set [18] respectively. The Office for National Statistics [19] supplied data on mortality among CO@h programme recipients. Descriptive data on participating CCGs in England were collected from a variety of sources, including departmental [20–22] and non-departmental [23–24] public bodies. Linkage across data-sets was achieved via a de-personalised (pseudonymised) NHS patient ID.

### Patient and public involvement

The study did not seek to incorporate the opinion of patient or public representatives in its design, analysis or interpretation of findings stages. This was largely due to the rapid nature of the evaluation, whose findings were needed to inform policy decision-making around the roll-out and effectiveness of the CO@h programme at a time of heightened public health urgency due to the spread of the COVID-19 pandemic.

### Results

In order to establish a reasonable separation between high- and low-onboarding CCGs, and concurrently identify the time frame during which to assess their emergency hospital use (the study follow-up period), onboarding fractions from participating CCGs were examined during the second wave of the pandemic.

Fig 1 displays median CO@h onboarding fractions over the first 18 weeks in 2021 (from 4 January throughout 9 May) for the 40 CCGs in England submitting complete or nearly complete onboarding data [7] to the CO@h programme. It should be noted that weekly data submissions to the programme do not allow discerning omitted from nil submissions. A number of features emerge from the distribution of CO@h onboarding fractions: notably, in this period the average median fraction (across participating CCGs) was only 4%. As previously mentioned, levels of programme uptake substantially below target posed a major challenge to the design and analysis of the evaluation: only 2,710 individuals (that is 2.12% of the eligible adult population) were ultimately onboarded onto a CO@h pathway between 4 January 2021 and 9 May 2021 across the above CCGs. Of note, the onboarded eligible proportion dropped to 0.92% when considering the 25,529

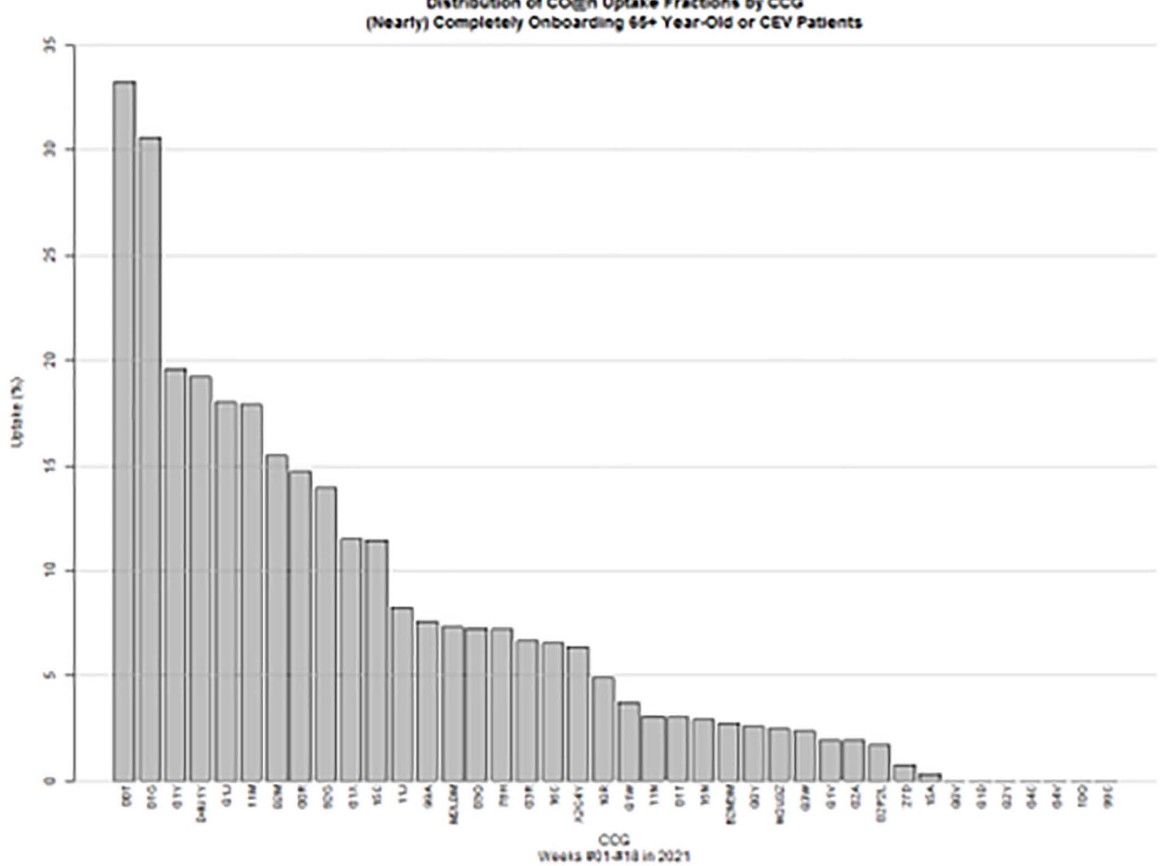

**Fig 1. CO@h uptake fractions.** Distribution of CO@h uptake fractions by.CCGs completely and nearly completely submitting patients returns to the CO@h programme.

eligible adults onboarded during the pandemic's second wave (conventionally set between 28 September 2020 and 04 July 2021).The study pre-intervention period, from which information on patient and CCG characteristics are sourced to inform the counterfactual-building process, was also gauged from AHSN return volumes to cover the 13 weeks prior to follow-up (5 October 2020–3 January 2021).

The average median onboarding fraction of 5.25% during follow-up, which is indicative of a considerable volatility in onboarding patterns both between CCGs as well as within CCGs over time, is markedly distant from the nominal 100% to be expected had the CO@h programme standard operating procedure been fully adhered to.

Finally, to ensure sufficient numbers of CCGs would inform the evaluation, from the distribution of onboarding fractions in Fig 1 the thresholds of 0.15 and 0.05 were judged to reasonably separate high from low CO@h onboarding fractions respectively. The adopted cut-off values not only retain in the analysis a degree of homogeneity in average programme uptake between contrasted groups of CCGs, as is apparent from the drops in uptake fraction around those values seen from Fig 1; they also allow informing the analysis with a sufficient number of comparison CCGs, as recommended for a reliable GSynth model fit [9]. This in turn led, after discarding 1 high-onboarding and 1 low-onboarding CCG affected by severe data completeness issues, to identifying an intervention and control group comprising respectively 6 (i.e., Namely Bolton CCG, Salford CCG, Tameside and Glossop CCG, Frimley CCG, Knowsley CCG and Gloucestershire CCG) and 20 CCGs.

Tables 1-2 display descriptive statistics respectively for the overall population of adults registered with a GP practice (Table 1) and for the patients on the CO@h pathway (Table 2) As to be expected in comparative effectiveness analyses based on observational data, significant differences in the distribution of most of the predictor and outcome variables were noted in the eligible population, both in the pre-intervention and follow-up periods, between the clusters of CCGs being compared, as well as with the remainder (excluded from the analysis). Differences at baseline in the general population at CCG cluster level aren't particularly surprising, given how broadly heterogeneous the groups of CCGs are in terms of socio-demographic and geographic characteristics. On the other hand, as also previously remarked the paucity of patients onboarded onto a weekly basis by participating CCGs is especially apparent, highlighting deviations from the programme's eligibility criteria at a local level. Notable differences between compared CCG clusters during the study follow-up largely concern emergency hospital activity rates and comorbidities – generally higher among low-onboarding CCGs – rather than socio-demographic characteristics. The choice of the GSynth approach to allow for impact estimation from the CO@h programme was largely made in anticipation of the presence of such observed and unmeasured discrepancies between comparison CCG groups, since the method is designed to account for these sorts of bias [9].

Fig 2 illustrates GSynth inferences, adjusted for the socio-economic, demographic and prognostic CCG-level characteristics listed in Tables (1)-(2), around the weekly effect of the CO@h programme among participating CCGs selected as high-onboarding, relative to those identified as low-onboarding, on each of the examined outcomes throughout the assumed pre-intervention and follow-up study periods. It can be noticed that no effect estimate is statistically significant for any of the outcomes during the pre-intervention period: this is to be expected, and is also regarded as a prerequisite indication of goodness of model fit to the data. Importantly, Fig 2 shows a near lack of statistically significant impact also throughout the follow-up period: the derivation of isolated statistically significant reductions in A&E attendance and emergency admissions rates in week #11 of 2021 is regarded as an artifact of the estimation process and interpreted as a null finding (type 1 statistical error). Of note, the widening over time of confidence bounds derived around the estimated effects is due to the increased sparsity of returns, as the CO@h programme progressively wound down. Overall (that is over the

**Table 1. Descriptive statistics for the population registered with a GP in CCGs included and excluded from the evaluation, expressed as weekly averages (percentages), over the pre-intervention period (05 October 2020–3 January 2021).**

| Variable | | High-onboarding CCGs | Low-onboarding CCGs | Other CCGs |
|---|---|---|---|---|
| CCG size | | 363,310.9 | 422,477.6 | 193,335.7 |
| Male | | 179,581.2 (49.4%) | 208,400.7 (49.3%) | 94,659.7 (49.0%) |
| Age | 18–24 | 28,951.8 (8.0%) | 37,279.0 (8.8%) | 13,083.4 (6.8%) |
| | 25–64 | 188,842.3 (52.0%) | 213,581.2 (50.6%) | 97,172.7 (50.3%) |
| | 65–74 | 34,893.5 (9.6%) | 45,046.7 (10.7%) | 23,333.6 (12.1%) |
| | 75 – | 29,016.6 (8.0%) | 38,719.5 (9.2%) | 20,277.3 (10.5%) |
| Ethnicity | White | 316,271.5 (87.1%) | 374,064.0 (88.5%) | 179,685.8 (92.9%) |
| | Black | 7,032.2 (1.9%) | 7,414.7 (1.8%) | 1,241.0 (0.6%) |
| | Asian | 30,995.4 (8.5%) | 29,915.8 (7.1%) | 8,918.9 (4.6%) |
| | Mixed | 6,633.1 (1.8%) | 7,574.4 (1.8%) | 2,566.5 (1.3%) |
| | Other | 2,378.2 (0.7%) | 3,508.8 (0.8%) | 923.5 (0.5%) |
| Educated to at least third level | | 44,071.3 (12.1%) | 54,958.3 (13.0%) | 23,727 (12.3%) |
| Population density (per square kilometre) | | 4,075.9 | 4,669.7 | 3,135.0 |
| Rural/ urban classification index | | 0.3 | 0.6 | 0.5 |
| Number of full-time-equivalent GPs (per 1,000 patients) | | 0.6 | 0.6 | 0.5 |
| 2019 Index of Multiple Deprivation | | 28.5 | 21.8 | 21.7 |
| 2019 Health Deprivation and Disability Index | | 0.5 | 0.1 | 0.3 |
| 2019 Income Deprivation Affecting Older People Index | | 0.2 | 0.2 | 0.1 |

**Table 2.** Descriptive statistics for the eligible population onboarded onto a CO@h clinical pathway in CCGs included and excluded from the evaluation, expressed as weekly averages (percentages), over the follow-up period (04 January 2021–09 May 2021).

| Variable | | High-onboarding CCGs | Low-onboarding CCGs | Other CCGs |
|---|---|---|---|---|
| Size | | 7.7 | 9.6 | 8.6 |
| Male | | 3.7 (47.6%) | 4.7 (48.9%) | 4.2 (48.4%) |
| Age | 18–24 | 0.4 (5.3%) | 0.7 (6.4%) | 0.3 (3.7%) |
| | 25–64 | 2.7 (35.0%) | 3.0 (31.6%) | 2.3 (28.4%) |
| | 65–74 | 2.4 (31.3%) | 3.0 (32.0%) | 3.0 (35.0%) |
| | 75 – | 2.2 (28.3%) | 2.9 (30.0%) | 2.9 (32.9%) |
| Ethnicity | White | 4.2 (56.1%) | 5.4 (58.3%) | 6.8 (75.1%) |
| | Black | 0.8 (9.8%) | 0.8 (8.2%) | 0.2 (3.6%) |
| | Asian | 1.8 (22.0%) | 2.0 (21.5%) | 1.0 (14.4%) |
| | Mixed | 0.4 (5.3%) | 0.6 (5.0%) | 0.0 (0.3%) |
| | Other | 0.6 (6.8%) | 0.8 (6.0%) | 0.5 (6.6%) |
| A&E attendances (per 100 patients) | | 41.7 | 53.9 | 42.0 |
| Emergency admissions (per 100 patients) | | 36.1 | 49.8 | 40.9 |
| Hospital bed days due to emergency admissions | | 562.5 | 711.2 | 663.9 |
| Critical care admissions (per 100 patients) | | 3.8 | 5.0 | 3.4 |
| Deaths (per 100 patients) | | 17.5 | 22.2 | 21.2 |
| Alcohol problems in 2 years prior to positive COVID-19 test | | 10.1 | 11.4 | 14.6 |
| Autism | | 0.3 | 0.6 | 0.4 |
| Chronic heart disease with atrial fibrillation | | 16.8 | 22.6 | 22.0 |
| Chronic kidney disease | | 6.6 | 8.9 | 7.7 |
| Chronic respiratory disease | | 5.0 | 8.3 | 7.4 |
| Asthma | | 23.4 | 33.6 | 29.4 |
| Chronic obstructive pulmonary disease | | 15.3 | 18.0 | 19.9 |
| Dementia | | 12.1 | 15.1 | 16.3 |
| Diabetes | | 40.7 | 60.3 | 44.0 |
| Flu | | 41.0 | 60.3 | 56.6 |
| Gestational diabetes | | 5.2 | 7.2 | 4.3 |
| Hypertension | | 64.2 | 93.1 | 80.8 |
| Hypothyroidism | | 15.8 | 22.7 | 19.8 |
| Learning disability | | 2.0 | 2.7 | 2.1 |
| Malignancy or immunosuppression | | 30.7 | 43.5 | 39.7 |
| Severe mental illness | | 2.7 | 4.2 | 3.7 |
| Palliative care | | 10.7 | 12.0 | 11.3 |
| Pre-diabetes | | 9.8 | 19.1 | 13.7 |
| Pregnancy | | 9.6 | 13.3 | 10.2 |
| Peripheral vascular disease | | 4.8 | 6.0 | 6.4 |
| Stroke | | 9.8 | 13.7 | 14.1 |
| Transient ischaemic attack | | 6.6 | 8.5 | 8.8 |
| Stage 4 chronic kidney disease | | 2.8 | 4.0 | 3.3 |
| Stage 5 chronic kidney disease | | 1.1 | 1.7 | 1.0 |
| Chronic kidney disease with dialysis | | 0.9 | 2.1 | 1.1 |
| Other chronic kidney disease | | 1.3 | 2.0 | 1.5 |
| Constipation | | 36.6 | 54.8 | 49.4 |
| Chronic respiratory disease with medication | | 42.6 | 59.4 | 53.9 |
| Diabetes not treated with Metformin | | 41.0 | 60.7 | 44.3 |

*(Continued)*

**Table 2.** (Continued)

| Variable | High-onboarding CCGs | Low-onboarding CCGs | Other CCGs |
|---|---|---|---|
| Immunosuppression | 12.7 | 18.5 | 16.0 |
| Hematologic malignancy | 4.1 | 6.0 | 5.3 |
| Solid malignancy | 16.9 | 23.2 | 22.9 |
| Sever mental illness under medication | 7.8 | 10.7 | 10.1 |
| Rheumatoid arthritis | 3.8 | 5.5 | 4.8 |
| Stroke and transient ischaemic attack | 13.7 | 19.1 | 19.3 |
| Diabetes treated with Metformin | 41.8 | 61.5 | 44.9 |
| Rheumatoid arthritis | 14.6 | 21.1 | 18.2 |
| Malignancy | 20.1 | 27.7 | 26.8 |

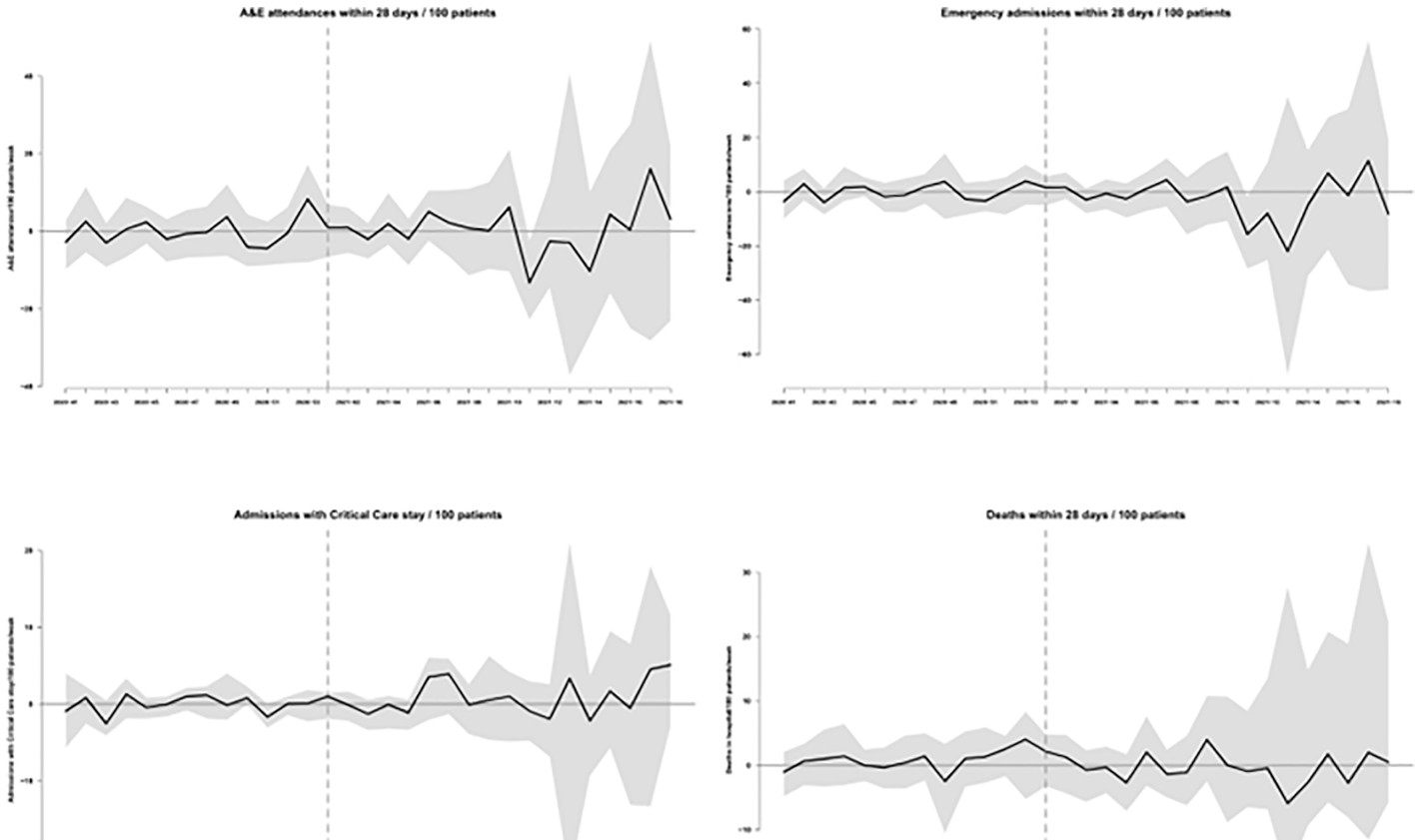

**Fig 2. Impact estimates.** Time-series plots of impact estimates (black solid line) and their 95% confidence intervals (shaded area) on rates of A&E attendance, emergency admission, admission into critical care and mortality. Weekly impact on a given outcome is derived from the difference in that week between the outcome rate among high-onboarding CCGs and its estimated counterfactual.

follow-up period) effect estimates showed no statistically significant impact for A&E attendances (estimated rate per 100 patients: 0.52; 95% confidence interval (CI): −7.81, 7.95), emergency admissions (estimated rate per 100 patients: −2.36; 95% CI: −14.54, 9.72), admissions with critical care stay (estimated rate per 100 patients: 0.91; 95% CI: −4.12, 3.66) and mortality (estimated rate per 100 patients: −0.29; 95% CI: −2.93, 8.66) within 28 days of a positive COVID-19 diagnosis. This null significance finding was confirmed also in 4 distinct scenario-based sensitivity analyses (results not shown), derived from altering the high and low onboarding fractions so that 4 and 9 CCGs would be retained as high-uptake and 16 and 28 as low-uptake in each respectively analysis.

## Discussion

The performed evaluation indicates that the CO@h programme didn't have a statistically significant impact on the selected emergency hospital activity indicators, as well as mortality, among CCGs in England consistently onboarding a high proportion of the eligible population (i.e., individuals of 65 years of age or over testing positive for COVID-19 or clinically extremely vulnerable adults), relative to those instead onboarding fewer cases, for much of the second wave of the COVID-19 pandemic. This key finding also appears to be broadly insensitive to how the groups of CCGs being compared were formed.

A number of plausible explanations for this conclusion can be devised, other than the programme lacking the anticipated effectiveness on the examined outcomes. Firstly, it must be stressed that enrolment onto the CO@h programme, which was intended to cover 100% of the eligible population, turned out to be unexpectedly low (2.12%). This proportion dropped to 0.92% when considering the 25,529 eligible adults onboarded during the pandemic's second wave (notionally between 28 September 2020 and 04 July 2021). Such low coverage is plausibly the result of a combination of incomplete and discontinuous returns on onboarded patients submitted to the programme by participating CCGs, and of genuinely few patient referrals onto a CO@h clinical pathway. Unfortunately, the coding of CO@h onboarding returns collected by the Kent, Surrey and Sussex AHSN does not allow distinguishing between nil and omitted CCG submissions, making it hard to discern the root causes of such low programme uptake across the country. In addition, anecdotal information shared in weekly sit rep programme meetings emphasised the inability by a number of participating CCGs in maintaining a reliable, if any, schedule of programme returns due to the severe strain they were operating under, while at the height of the pandemic outbreak. Moreover, information governance restrictions set by electronic data infrastructure providers for several participating CCGs raised long-standing barriers to programme data access that couldn't be timely resolved. Notwithstanding the above considerations, we have no knowledge of there being reasons linked to local programme onboarding or delivery practices or patterns that might have selectively led CCGs to withhold onboarding data, and hence induced selection bias in our evaluation findings.

Secondly, and in relation with such low programme uptake, is the possibility that any effect on clinical outcomes conveyed by the available data is, at least in part, a reflection of health-care transformation initiatives that are independent of the CO@h programme. Among such initiatives known to have taken place concurrently or around the time CO@h was implemented are the roll-out of the COVID-19 vaccination programme, the introduction of COVID virtual wards and the PRINCIPLE trial. The NHS began administering COVID-19 vaccinations in England on 8 December 2020 with the primary aims of preventing COVID-19 mortality and protecting health and social care staff and systems, and the secondary aims of protecting individuals at increased risk of hospitalisation or infection and maintaining resilience in essential public services [25]. Accordingly, the national roll-out of COVID-19 vaccinations was staggered across the population, prioritising individuals in older age groups and with serious underlying health conditions. COVID virtual wards, a complementary but separate programme to CO@h put into operation in England at the end of December 2020, was aimed at promoting early supported hospital discharge and safe admission avoidance (including with, but not limited to, the aid of pulse oximeters) for patients already hospitalised with COVID-19 [26]. The PRINCIPLE study is a national priority randomised trial set in primary care, evaluating usual care alone versus a combination of usual care and different antiviral or antiparasitic drugs,

with the purpose of investigating safe treatment at home of COVID-19. The trial began on 17 April 2020 by enrolling adults showing symptoms of, or testing positive for, COVID-19 in the previous 14 days; it reached 4,053 participants on 11 February 2021 [27]. None of the aforementioned COVID-19 mitigation initiatives fall within the scope of the outlined evaluation of the CO@h programme; nevertheless their influence on clinical outcomes hereby examined cannot be conclusively ruled out, or discerned from available data.

Thirdly, it should be kept in mind that the proposed evaluation only detects observed differences between the compared CCG clusters. Potentially important confounders, like the quality of care provided and the severity of pre-existing conditions afflicting onboarded patients, are not routinely recorded in the administrative health data-bases informing this study, but may still be partly responsible for differences between high- and low-onboarding CCGs that remain unaccounted for by this evaluation.

Fourthly, as raised by a peer-reviewer programme fidelity may have played a role in affecting the relationship between the CO@h intervention and the outcomes we have examined. As described in the Introduction the programme was delivered across onboarding CCGs with a high degree of heterogeneity both in mode of delivery and eligibility criteria (*fidelity*). By design, eligible patients in CCGs participating to the programme alongside our study follow-up period were either offered the CO@h pathway or not (exposure). We had no access throughout the analysis to information relating to delivery of the CO@h programme to its recipients (quality of delivery) or to participant engagement (adherence), though a separate study uncovered a broadly positive reception of the intervention by patients and staff despite some engagement and operationalisation challenges [6]. We have previously commented notably on the introduction in England of COVID virtual wards which, while different from the CO@h programme in aims and scope, largely overlapped with it in delivery timeline and presented similarities in its implementation (programme differentiation). Of note, we never had access to data around, or any involvement with, the COVID virtual ward programme. All in all we would conclude that, while imperfect, programme fidelity would have unlikely affected our findings to a substantial enough extent as to obfuscate the intervention – outcomes relationship our evaluation failed to discern (assuming it was present).

The proposed impact evaluation relies on a formal statistical modelling framework (GSynth) recognised for its flexibility, robustness and ability to capture time-varying and/ or unobserved effects (e.g., disease severity at diagnosis) from available data. Each outcome model was fitted to a complex network of registry and surveillance data sources, linked to primary and secondary administrative records, forming a uniquely comprehensive evidence base specifically around the population targeted by the CO@h programme. On the other hand, the previously described unforeseen limitations in data volumes, quality and completeness, exacerbated by local variation in the implementation of the pathway from the programme's suggested standard operating procedure, cast a doubt on the evaluation's adequacy to discern a link (if existing) between the effectiveness of CO@h clinical pathways and CCG-level onboarding volumes. As is typically the case with comparative effectiveness research based on retrospectively collected, observational data, we utilised as much data were viably made available to us both from CCGs participating and not yet participating into the programme. As such, no power calculation exercise was carried out at the design stage of the proposed evaluation to discern minimal sample size requirements. In retrospect, such an exercise would have proved futile given the severely low programme enrolment our evaluation had to grapple with.

Alternative impact evaluation designs (in particular an individual patient-level analysis based on a Regression Discontinuity Design approach [28]) were rendered unfeasible by the above data sparsity and protocol adherence issues and required falling back to addressing an arguably less informative and generalisable, if equally valid, hypothesis. The same severe data limitations, combined with the CCG-level approach to the evaluation, prevented us in practice from carrying out subgroup analyses (for instance focussing on patients from an ethnic minority background), which an individual-level analysis would have been better suited for.

It is worth noting that the evidence conveyed by the present evaluation fits broadly with the body of evidence developing around not only the effectiveness of the CO@h programme, but also the wider benefits of the use of pulse oximeters

for remote home monitoring of hypoxia. A stepped wedge post- vs pre-intervention approach to the same data sources found no meaningful change in mortality at CCG level following CO@h implementation and small, if statistically significant, increases in the odds of A&E attendance and of emergency admission [29]. Another population-level dose-response type assessment of the clinical effectiveness of the CO@h programme, also largely overlapping with that hereby illustrated in target population, underlying data, outcome metrics and time-frames, uncovered no statistically significant association between CO@h territorial coverage and emergency admissions, mortality and length of hospital stay [30]. Furthermore, a national analysis of essentially the same cohort targeted by the present evaluation found that COVID-19 hospitalisations and mortality rose significantly over time in England during the second pandemic wave [31]. Given the descriptive (observational and non-comparative) nature of this study, care has to be taken in the interpretation of these findings as they may be plausibly attributed to a variety of factors including hospital and winter pressures, changes in admission thresholds, a heightened awareness of the negative effects of silent hypoxia and the concurrent spread of the highly morbid Alpha COVID-19 strain. Of note, being our study an impact evaluation its target estimands consisted of the Average Treatment Effects on the Treated (ATT); not on quantifying the influence of included covariates (and interactions thereof) on the outcomes of interest, which may be explored via a more aptly designed study ideally informed by a better powered sample.

Finally, local variations in the implementation of the CO@h pathways from the standard operating procedures set by the programme should not be seen as a criticism of frontline teams delivering the service. Given the time sensitivity and overall pressure on the health care system, many of the CO@h pathways were developed locally either ahead of (in the case of some pilot CCGs) or alongside the development of national (suggested) standard operating procedures. The national programme aimed for a service allowing patients to self-monitor with little clinical input, whereas in practice many CCGs implemented a remote monitoring service requiring high staffing level, leading to reduced capacity and an overall lower level of onboarding. What capacity was available was targeted at those patients who would likely benefit the most, which in turn resulted in variations in eligibility criteria where clinical judgement plays a bigger role [25].

## Conclusions

Ultimately there are a number of possible reasons why the present evaluation, in combination with the above reviewed independent studies, essentially failed to detect a statistically meaningful impact for the CO@h programme. Firstly, CO@h may have simply not significantly affected COVID-19 emergency hospital resource utilisation and mortality in high-onboarding CCGs compared to low-onboarding CCGs. Secondly, unforeseen low onboarding levels, coupled with incompleteness and sparsity of programme data across England, may have compromised the ability of the adopted evaluation strategy to discern an existing impact. Thirdly, the existence of various modes (pre-hospital or post-discharge) and models (primary care, secondary care, step-down, mixed) of programme implementation across the country may have led to diluting the overall programme effectiveness. Fourthly, associations existing at individual patient level may be eluded by aggregate-level analyses. Importantly, a qualitative assessment of the processes of CO@h implementation and a survey of patient and staff experiences suggested that the programme was in fact well received by certain population subgroups (patients, health-care staff and programme workers), although challenges were noted around engagement with the service and the availability of operational support [6].

## Data sharing statement

There are legal restrictions on the data. The authors gained access to the data through Control of Patient Information (COPI) notices issued by the Department of Health and Social Care under Regulation 3(4) of the Health Service Control of Patient Information Regulations 2002, which required organizations to share confidential patient information relating to COVID-19 with select organizations. COPI Notices expired as of 30 June 2022. Accordingly the full dataset informing this study had to be contextually deleted from the secure data environment it was stored in, and made accessible to, the Improvement Analytics Unit.

In the event of an alternative lawful basis being identified that would permit continued processing of confidential patient information, applicants can contact the NHS Health Research Authority Confidentiality Advisory Group (URL: https://www.hra.nhs.uk/about-us/committees-and-services/confidentiality-advisory-group; e-mail: cag@hra.nhs.uk): an independent body providing expert advice on the use of confidential patient information whose mission is to protect and promote the interests of patients and the public in the UK, while at the same time facilitating appropriate use of confidential patient information for purposes beyond direct patient care.

**Strengths and limitations of this study**

- The study is an original evaluation of the impact of the COVID Oximetry @home national programme on a range of hospital activity and mortality indicators in England between high- and low-onboarding CCGs during the second wave of the COVID-19 pandemic.

- Impact quantification is based on a rigorous and flexible causal modelling framework, which allows for unobserved and time-varying confounding.

- Sensitivity analysis showed robustness of main analysis findings to adopted high- and low-onboarding CCG thresholds.

- Significant issues with data collection and patient onboarding rates at different CCGs hampered the study's ability to detect an impact (if present).

- Data sparsity issues compounded with lack of uniformity in programme implementation across onboarding CCGs to hinder adoption of alternative causal modelling frameworks (e.g., a Regression Discontinuity Design approach)

**Key messages**

**What is already known on this topic**

The COVID Oximetry @home (CO@h) programme was rolled out in November 2020 across England to support self-management of COVID-19 patients at risk of health deterioration (due to silent hypoxia) via the provision of pulse oximeters.

**What this study adds**

This study presents the findings from a quantitative evaluation of the impact of this programme on emergency hospital use and mortality. No statistically significant impact was detected on selected emergency hospital activity indicators and mortality among CCGs in England consistently onboarding a high proportion of the eligible population, relative to those instead onboarding fewer cases, for much of the second wave of the COVID-19 pandemic.

**How this study might affect research, practice or policy**

Our inability to recover statistically significant evidence of impact of the CO@h programme on emergency hospital utilisation or mortality may be due to an absence of impact, or a combination of unexpectedly low levels of uptake across participating areas and a high degree of heterogeneity in implementation across sites. Standard operating procedures were issued as guidance only, with many local sites already delivering a different clinical model.

**Supporting information**

**S1 File. COPI Notice.** Notice under Regulation 3(4) of the Health Service Control of Patient Information Regulations 2002. (DOCX)

## Author contributions

**Conceptualization:** Stefano Conti, Paris Pariza, Arne Wolters.

**Data curation:** Paris Pariza.

**Formal analysis:** Stefano Conti, Paris Pariza.

**Investigation:** Arne Wolters.

**Methodology:** Stefano Conti, Arne Wolters.

**Project administration:** Stefano Conti, Arne Wolters.

**Resources:** Arne Wolters.

**Software:** Paris Pariza.

**Supervision:** Stefano Conti, Arne Wolters.

**Validation:** Stefano Conti, Paris Pariza.

**Visualization:** Paris Pariza.

**Writing – original draft:** Stefano Conti, Arne Wolters.

**Writing – review & editing:** Stefano Conti, Paris Pariza, Arne Wolters.

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
