## [Decision Letter · Decision Letter 0]

16 Aug 2024

Dear Dr. Conti,

Thank you for submitting your manuscript to PLOS ONE. After careful consideration, we feel that it has merit but does not fully meet PLOS ONE’s publication criteria as it currently stands. Therefore, we invite you to submit a revised version of the manuscript that addresses the points raised during the review process.

We look forward to receiving your revised manuscript.

Kind regards,

Pisirai Ndarukwa, Ph.D.

Academic Editor

PLOS ONE

Reviewers' comments:

Reviewer's Responses to Questions

**Comments to the Author**

1. Is the manuscript technically sound, and do the data support the conclusions?

Reviewer #1: Yes

2. Has the statistical analysis been performed appropriately and rigorously?

Reviewer #1: Yes

3. Have the authors made all data underlying the findings in their manuscript fully available?

Reviewer #1: No

4. Is the manuscript presented in an intelligible fashion and written in standard English?

Reviewer #1: Yes

Reviewer #1: I find this a clearly presented study. Although the COVID pandemic is over, this has relevance to evaluation of virtual wards in general, which are becoming more common across the NHS and for which robust evaluation is limited.

**Do you want your identity to be public for this peer review?** For information about this choice, including consent withdrawal, please see our Privacy Policy

Reviewer #1: No

---

## [Author Response · Author response to Decision Letter 1]

26 Sep 2024

We would like to acknowledge the Associate Editor and peer-reviewers for taking their time and offer their considerations to improve and strengthen our submission. We have considered all provided comments and included revisions to the original manuscript to address them for further consideration.

We have provided our response below and uploaded the revised manuscript (clean and edited versions) for further consideration.

Yours sincerely,

Dr Stefano Conti (on behalf of all authors)

Reviewer #1

[#3]

Like we explained in a separate addendum to the original submission, unfortunately we are unable to share even a minimal version of the data-set (which in fact is no longer available to us) underpinning our study. According to guidelines issued by the NHS Health Research Authority, since the submitted manuscript describes a service evaluation it requires no NHS Research Ethics Committee review.

More in detail, on April the 1st 2020 the United Kingdom's Secretary of State for Health and Social Care issued Notices under Regulation 3(4) of the Health Service (Control of Patient Information) Regulations 2002 (COPI), which required organisations to share confidential patient information with organisations – including the Improvement Analytics Unit, which all authors were part of at the time of this study being carried out – entitled to process this under COPI for COVID-19 purposes (COPI Notices). We have received anonymised patient-level data to inform the impact evaluation outlined in this submission under the above COPI Notices, which have provided us with the necessary and sufficient legal basis for the processing of the confidential patient information underpinning our evaluation that was required for managing the response to the COVID-19 pandemic. Accordingly, in the light of the guidance for organisations on processing of confidential patient information under the COPI Notices, no NHS Research Ethics Committee review of our study was required by our study. Of note, the COPI Notices expired of June the 22nd 2022.

We hope for the above information to satisfy the peer-reviewer’s concern about the lacking provision of the data-set informing our evaluation. To this end we revised the original Data Sharing Statement section to convey the above information.

[#5]

We thank the peer-reviewer on her / his appreciation of the potential value of our work in the perspective of the developing Virtual Wards programme, which the Improvement Analytics Unit are also separately researching.

---

## [Decision Letter · Decision Letter 1]

31 Mar 2025

Dear Dr. Conti,

Thank you for submitting your manuscript to PLOS ONE. After careful consideration, we feel that it has merit but does not fully meet PLOS ONE’s publication criteria as it currently stands. Therefore, we invite you to submit a revised version of the manuscript that addresses the points raised during the review process.

**ACADEMIC EDITOR:** The manuscript has received generally positive feedback. Reviewer 1 indicates that all previous comments have been satisfactorily addressed and that the study is technically sound, with appropriate data availability and presentation. However, Reviewer 2 has raised several concerns that warrant minor revisions. In particular, the following issues should be addressed:

• Counterfactual Validation: Please provide further details on any placebo or permutation tests you have conducted to support the accuracy of the Generalized Synthetic Control (GSynth) approach. If such tests have not been performed, a discussion of the potential limitations and planned future validations would be beneficial.

• Sensitivity to Alternative Methods: It would strengthen the manuscript to include a brief sensitivity analysis or discussion comparing the GSynth results with alternative causal inference methods such as Difference-in-Differences, Bayesian models, or Regression Discontinuity designs. This would help assess the robustness of your findings in light of the low program uptake.

• Data Sparsity and Bias: Given that the program uptake was very low (approximately 2.12%), please clarify how you ensured that the counterfactual comparisons were not biased by data sparsity. Additional commentary on the limitations imposed by these data issues is encouraged.

• Interpretability of Results: The reviewer suggested exploring methods such as Partial Dependence Plots (PDPs) or SHAP values to better understand the interactions between patient characteristics and outcomes. While not essential, any additional insights or discussion on the interpretability of the GSynth findings would add value to your analysis.

We believe that addressing these points in a revised version will further improve the robustness and clarity of your work. Once these revisions have been made and adequately described, the manuscript should be suitable for publication.

We look forward to receiving your revised manuscript.

Kind regards,

Mohsen Mehrabi

Academic Editor

PLOS ONE

Journal Requirements:

Additional Editor Comments:

The manuscript has received generally positive feedback. Reviewer 1 indicates that all previous comments have been satisfactorily addressed and that the study is technically sound, with appropriate data availability and presentation. However, Reviewer 2 has raised several concerns that warrant minor revisions. In particular, the following issues should be addressed:

• Counterfactual Validation: Please provide further details on any placebo or permutation tests you have conducted to support the accuracy of the Generalized Synthetic Control (GSynth) approach. If such tests have not been performed, a discussion of the potential limitations and planned future validations would be beneficial.

• Sensitivity to Alternative Methods: It would strengthen the manuscript to include a brief sensitivity analysis or discussion comparing the GSynth results with alternative causal inference methods such as Difference-in-Differences, Bayesian models, or Regression Discontinuity designs. This would help assess the robustness of your findings in light of the low program uptake.

• Data Sparsity and Bias: Given that the program uptake was very low (approximately 2.12%), please clarify how you ensured that the counterfactual comparisons were not biased by data sparsity. Additional commentary on the limitations imposed by these data issues is encouraged.

• Interpretability of Results: The reviewer suggested exploring methods such as Partial Dependence Plots (PDPs) or SHAP values to better understand the interactions between patient characteristics and outcomes. While not essential, any additional insights or discussion on the interpretability of the GSynth findings would add value to your analysis.

We believe that addressing these points in a revised version will further improve the robustness and clarity of your work. Once these revisions have been made and adequately described, the manuscript should be suitable for publication.

Reviewers' comments:

Reviewer's Responses to Questions

**Comments to the Author**

Reviewer #1: All comments have been addressed

Reviewer #2: (No Response)

2. Is the manuscript technically sound, and do the data support the conclusions?

Reviewer #1: Yes

Reviewer #2: Partly

3. Has the statistical analysis been performed appropriately and rigorously?

Reviewer #1: Yes

Reviewer #2: No

4. Have the authors made all data underlying the findings in their manuscript fully available?

Reviewer #1: Yes

Reviewer #2: No

5. Is the manuscript presented in an intelligible fashion and written in standard English?

Reviewer #1: Yes

Reviewer #2: Yes

Reviewer #1: The authors have adequately addressed the question about access to data. I have no further comments.

Reviewer #2: Thank you for your valuable evaluation of the COVID Oximetry @home (CO@h). While your use of Generalized Synthetic Control (GSynth) is commendable, we have questions regarding counterfactual validation and interpretability.

Counterfactual Validation: Did you conduct placebo tests or permutation tests to confirm GSynth’s accuracy? How sensitive are your results to alternative methods such as Difference-in-Differences (DiD), Bayesian models, or Regression Discontinuity (RD)?

Data Sparsity Issues: Given the low program uptake (2.12%), how did you ensure unbiased counterfactual comparisons?

Interpretability: Did you explore Partial Dependence Plots (PDPs) or SHAP values to assess interactions between patient characteristics and outcomes? Machine learning models could provide deeper insights.

Aggregate vs. Patient-Level Analysis: Would an RD approach at the patient level (e.g., using age eligibility) yield clearer causal effects? How did you account for CCG-level implementation heterogeneity?

To strengthen the findings, we recommend counterfactual robustness checks, alternative causal methods, and improved interpretability analysis. Looking forward to your response.

Overall a great job but could be strengthen with updated software packages and methods.

**Do you want your identity to be public for this peer review?** For information about this choice, including consent withdrawal, please see our Privacy Policy

Reviewer #1: No

Reviewer #2: **Yes: ** Taposh Dutta Roy

---

## [Author Response · Author response to Decision Letter 2]

2 Jun 2025

We thank Peer-Reviewer for the helpful feedback and suggestions she / he has produced, aimed at strengthening still under-developed aspects of our manuscript. We believe to have now addressed those considerations in the revised manuscript as recommended by the managing Editor in her 09.05.25 e-mail.

---

## [Decision Letter · Decision Letter 2]

30 Jun 2025

Dear Dr. Conti,

Thank you for submitting your manuscript to PLOS ONE. After careful consideration, we feel that it has merit but does not fully meet PLOS ONE’s publication criteria as it currently stands. Therefore, we invite you to submit a revised version of the manuscript that addresses the points raised during the review process.

We look forward to receiving your revised manuscript.

Kind regards,

Mohsen Mehrabi, Ph.D.

Academic Editor

PLOS ONE

Journal Requirements:

Reviewers' comments:

Reviewer's Responses to Questions

**Comments to the Author**

Reviewer #2: (No Response)

2. Is the manuscript technically sound, and do the data support the conclusions?

Reviewer #2: Yes

3. Has the statistical analysis been performed appropriately and rigorously?

Reviewer #2: Yes

4. Have the authors made all data underlying the findings in their manuscript fully available?

Reviewer #2: No

5. Is the manuscript presented in an intelligible fashion and written in standard English?

Reviewer #2: Yes

Reviewer #2: This manuscript presents a well-executed evaluation of the NHS England COVID Oximetry @home (CO@h) programme using a Generalized Synthetic Control (GSynth) method to assess its impact on emergency hospital utilization and mortality during the second wave of the COVID-19 pandemic. The authors leverage linked national datasets and appropriately acknowledge the limitations of their approach, including low programme uptake, heterogeneity in implementation, and data sparsity.

The methodology is robust and appropriate for the research question, and the discussion is balanced and transparent. However, the null findings—while possibly accurate—are difficult to interpret definitively given the limited onboarding rates and challenges with implementation fidelity across sites.

To strengthen the manuscript for publication, I recommend the following:

Clarify onboarding thresholds and their empirical justification.

Address selection bias from exclusion of CCGs with incomplete data.

Incorporate diagnostics for GSynth model fit and placebo tests in the supplement.

Add implementation fidelity measures (if available) or conduct subgroup analyses by delivery model.

Provide a more structured rationale for the observed null effects, possibly using power calculations or comparisons to other concurrent interventions (e.g., COVID Virtual Wards).

Despite limitations, this study contributes valuable insight into the complexities of evaluating national remote monitoring programmes under real-world conditions.

**Do you want your identity to be public for this peer review?** For information about this choice, including consent withdrawal, please see our Privacy Policy

Reviewer #2: **Yes: ** Taposh Dutta Roy

---

## [Author Response · Author response to Decision Letter 3]

14 Oct 2025

Please refer to the separately enclosed "Response to Reviewers 06.10.25" for details of revision work carried out on the previous version of the draft manuscript.

---

## [Decision Letter · Decision Letter 3]

12 Nov 2025

A synthetic control evaluation of the use of pulse oximeters in response to the COVID-19 pandemic in England

PONE-D-24-21517R3

Dear Dr. Conti,

We’re pleased to inform you that your manuscript has been judged scientifically suitable for publication and will be formally accepted for publication once it meets all outstanding technical requirements.

Kind regards,

Mohsen Mehrabi, Ph.D.

Academic Editor

PLOS ONE

Additional Editor Comments (optional):

Thank you for your thorough revisions to the manuscript titled "A synthetic control evaluation of the use of pulse oximeters in response to the COVID-19 pandemic in England." The manuscript has significantly improved, and both reviewers have expressed positive feedback.

Reviewers have commended the manuscript for its methodological rigor, transparency, and meaningful contribution to public health evaluations. They have suggested a few minor editorial adjustments to improve language flow, figure readability, and contextualization, among others. These suggestions are editorial in nature and are not critical to the scientific validity of the paper.

Based on the positive evaluations and the substantial revisions made, my final recommendation is to accept the manuscript for publication, contingent upon the authors making the minor editorial revisions as suggested by Reviewer 2. Once these adjustments are completed, the paper will be ready for publication.

Reviewers' comments:

Reviewer's Responses to Questions

**Comments to the Author**

Reviewer #2: All comments have been addressed

2. Is the manuscript technically sound, and do the data support the conclusions?

Reviewer #2: Yes

3. Has the statistical analysis been performed appropriately and rigorously?

Reviewer #2: Yes

4. Have the authors made all data underlying the findings in their manuscript fully available?

Reviewer #2: Yes

5. Is the manuscript presented in an intelligible fashion and written in standard English?

Reviewer #2: Yes

Reviewer #2: I have carefully reviewed your revised manuscript titled “A synthetic control evaluation of the use of pulse oximeters in response to the COVID-19 pandemic in England” (PONE-D-24-21517R3). I commend you for the thorough revisions and the clear, methodologically robust evaluation that now fully aligns with the scope and publication criteria of PLOS ONE. Below are my detailed comments.

Overall Assessment

This is a well-conceived, well-executed, and timely study that provides valuable empirical evidence on the impact of the NHS COVID Oximetry @home (CO@h) programme using a rigorous causal-inference framework. The paper addresses a major public-health and service-evaluation question with transparency and methodological integrity. Although the main findings are null, they remain highly informative for understanding real-world implementation challenges in large-scale digital health interventions during the COVID-19 pandemic.

The manuscript demonstrates a high level of analytical and ethical rigor, clear organization, and balanced interpretation of results. I believe this paper makes a meaningful contribution to the literature on population-level service evaluations, remote monitoring, and causal inference in public-health contexts.

Major Strengths

Methodological Appropriateness and Rigor

The use of the generalized synthetic control (GSynth) method is highly suitable for this policy evaluation. It effectively accounts for unobserved, time-varying confounding and provides an analytically sound counterfactual framework. The authors also validate their models using appropriate diagnostics (placebo tests, BIC, and mean squared prediction error), demonstrating careful implementation.

Transparency and Reflexivity

The manuscript is notably transparent about data limitations, low onboarding rates, and variation in programme implementation. The authors are candid about the structural and operational barriers—such as data sparsity, heterogeneous local practices, and legal constraints following the expiry of COPI notices—that limited their ability to detect statistically significant effects. This transparency strengthens the paper’s credibility.

Balanced Interpretation of Null Findings

The discussion thoughtfully situates the null results in context. The authors consider alternative explanations, such as concurrent interventions (vaccination rollout, COVID virtual wards, PRINCIPLE trial), and the challenges of data completeness. Importantly, they avoid overstating the findings while still highlighting the broader implications for healthcare evaluation methodology and policy.

Contribution to Evidence-Based Policymaking

The study’s findings have clear policy relevance. Demonstrating that large-scale monitoring programmes may not yield measurable system-level effects under conditions of low uptake and heterogeneous implementation offers an important lesson for the design and scaling of future digital-health initiatives.

Ethical and Data Governance Compliance

The paper provides a detailed and well-reasoned explanation of the ethical and legal framework (COPI Notices) under which patient-level data were accessed and processed. The justification for data unavailability post-June 2022 is fully compliant with NHS data governance standards and PLOS’s transparency requirements.

Minor Comments and Suggestions

Editorial Refinements

Consider a brief language polish for conciseness and flow, especially in the Discussion section where sentences could be simplified without losing nuance.

Ensure uniform tense usage when referring to the analysis (“was performed” vs. “is performed”) and standardize acronyms on first use.

Figures and Tables

Figures 1 and 2 are informative but could benefit from slightly improved readability (e.g., larger axis labels, consistent colour scheme).

Consider including a short summary line or caption note emphasizing that the confidence intervals overlap zero, reinforcing the null effect visually.

Contextualization of Related Studies

The references to independent evaluations (Beaney et al., Sherlaw-Johnson et al.) are well chosen. A single sentence highlighting how this work extends or differs from those analyses—for instance, by applying GSynth rather than stepped-wedge or regression approaches—would clarify the unique contribution.

Discussion on Programme Fidelity

The discussion around fidelity is balanced; however, a brief elaboration on how fidelity variation could inform future evaluation design (e.g., need for real-time standardization or adaptive monitoring) might add a practical perspective.

Patient and Public Involvement

Although the absence of PPI is justified due to pandemic urgency, you might include a short statement acknowledging how future evaluations could incorporate patient or public input once the immediate crisis phase has passed.

Conclusion Emphasis

The conclusion is strong, but consider ending with a sentence reinforcing the methodological value of this approach (“This evaluation illustrates how causal inference methods such as GSynth can support rapid policy learning in public-health emergencies, even when results are null.”).

These refinements are optional and editorial in nature; none are essential to publication.

Ethical and Publication Considerations

I find no concerns regarding dual publication, research ethics, or conflicts of interest. The ethics statement is comprehensive and appropriately references the NHS Health Research Authority guidance and COPI regulatory context. The data-availability statement is compliant and justified. There is no evidence of overlapping publication or prior dissemination that would violate PLOS ONE’s policies.

Final Recommendation

In my view, the manuscript now satisfies PLOS ONE’s standards for:

Scientific validity and methodological soundness

Transparency and reproducibility

Ethical compliance and data governance

Clear contribution to the evidence base for health-policy evaluation

The paper is ready for publication pending only minor editorial adjustments (e.g., copyediting and figure formatting). The authors have responded thoroughly to prior feedback, and no further analysis is required.

**Do you want your identity to be public for this peer review?** For information about this choice, including consent withdrawal, please see our Privacy Policy

Reviewer #2: **Yes: ** Taposh Dutta Roy

---

## [Editor Report · Acceptance letter]

PONE-D-24-21517R3

PLOS ONE

Dear Dr. Conti,

I'm pleased to inform you that your manuscript has been deemed suitable for publication in PLOS ONE. Congratulations! Your manuscript is now being handed over to our production team.

Kind regards,

on behalf of

Dr. Mohsen Mehrabi

Academic Editor

PLOS ONE